# Real-World Functioning in Psychiatric Outpatients: Predictive Factors

**DOI:** 10.3390/jcm11154400

**Published:** 2022-07-28

**Authors:** Paola Bozzatello, Benedetta Giordano, Cristiana Montemagni, Paola Rocca, Silvio Bellino

**Affiliations:** Department of Neuroscience, University of Turin, 10126 Turin, Italy; benedetta.giordano@unito.it (B.G.); cristiana.montemagni@unito.it (C.M.); paola.rocca@unito.it (P.R.); silvio.bellino@unito.it (S.B.)

**Keywords:** psychiatric disorders, outpatients, schizophrenia, bipolar disorder, major depression, borderline personality disorder, real-world functioning

## Abstract

Introduction: Investigations on predictors of real-world functioning were mainly performed in patients with schizophrenia, while fewer studies have been conducted in other psychiatric disorders. Objective: Our objective was to identify clinical, socio-demographic, and illness-related predictors of real-world functioning during 12 months of standard treatments in outpatients with different diagnoses. Methods: Outpatients (*n* = 1019) with schizophrenia (SZ), major depressive disorder (MDD), bipolar disorder (BD), and borderline personality disorder (BPD) were evaluated with the following tools: SCID-5-CV and SCID-5-PD, CGI-S, SAT-P, DAI-10, and PSP. Change of PSP (ΔPSP) between baseline and 12 months was used as the dependent variable in multiple regression analysis. Results: Higher PSP score at baseline and the achievement of main milestones predicted better functioning after follow-up in all subgroups of patients, with the exception of BD. In the total sample, ΔPSP was related to age of onset, treatments, and quality of life, and inversely related to psychiatric anamnesis, antidepressants, and global symptoms. In SZ, ΔPSP was related to adherence and quality of life. In MDD, ΔPSP was related to psychotherapy and quality of life, and inversely related to antidepressants and global symptoms. In BD, ΔPSP was related to age of onset, antipsychotics, and quality of life, while it was inversely related to psychiatric anamnesis. In BPD, antipsychotics, mood stabilizers, psychotherapy, and quality of life were directly related to ΔPSP, while suicidal attempts and global symptoms had an inverse relation. **Conclusions:** Several socio-demographic and illness-related variables predicted improvement of real-world functioning, besides psychopathology and severity of the disease.

## 1. Introduction

Despite last years’ significant advances in pharmacological and non-pharmacological treatments in the field of mental illness that have facilitated symptomatic remission, real-life disability produced by these diseases remains a huge burden for patients and their families in terms of health economic resources and days away from work. For this reason, the authors focused their investigations on functional outcomes, in particular on real-world functioning, which still represents an unmet need in people suffering from severe psychiatric disorders. with regard to functionality, two core concepts can be distinguished: (1) functional capacity (“what a patient can do”: the ability to obtain a good level of functioning under optimal conditions, which can be evaluated by performance-based ratings carried out in a neutral environment) and (2) real-world functioning (“what a patient actually does”: the actual performance when the subject faces the real circumstances of his life, assessed through measures carried out in the real-world) [1,2,3,4]. The everyday performance includes ordinary activities such as global organization, economic management, communication abilities and social interactions, independent living, and medication management. It has been noted that there is a gap between functional capacity and actual performance: functional capacity seems to be a predictor of real-world functioning (even though its effect is mediated by many other factors that influence behaviors in everyday life) and can contribute to the gap, particularly in schizophrenia (SZ) [3,4,5,6].

Most of the studies in this field are performed in patients with a diagnosis of schizophrenia as it produces a high degree of impairment in different domains of everyday life (employment, independent living, marital status, and interpersonal relationships) [7,8,9]. In this disorder, improvement of functioning in the real context represents the major objective of integrated treatments [10,11]. Several studies aimed at identifying the determinants of functioning stated that obtainment of real-life functional milestones, such as educational level, independent living, work, marital status, and interpersonal relationships, depends on a large variety of factors related to the characteristics of the illness, the individual, resources, and the social context of the patient [6,11,12,13,14,15,16,17,18,19,20]. In particular, recent meta-analyses highlighted the central role of neuro and social cognition, metacognitive abilities, and negative symptoms [21,22,23].

Over the years, studies on this topic have expanded to include other mental disorders, with a growing interest for bipolar disorder (BP). These patients showed a lower rate of achievement of functional milestones than the general population due to the long-lasting course of the disorder and the impairment in physical, work, and social functioning persisting long after symptomatic recovery, with an important reduction of patients’ quality of life (QoL) [24,25,26,27,28]. Several authors observed that mood symptoms and personal-related factors had a significant effect on a wide range of everyday expressions of functioning [27,29,30,31,32,33]. All phases of the disease, including symptom-free intervals, seem to have a role in compromising community functioning [34,35,36]. In addition, authors found that depressive phases are more damaging to real-life functioning than mania [37,38]. Moreover, the presence of subsyndromal depressive symptoms represents one of the main predictors of functional impairment, in particular of attitudes such as giving up and self-blame [32,39,40,41]. Recent papers focused specifically on neurocognitive function [27,28,30,42,43], as cognitive deficits were considered one of the main determinants of functional impairment, and evidence showed that better performance in some areas, such as attention, executive functions, and verbal memory, are predictors of better psychosocial functioning [44].

Even if major depressive disorder (MDD) is considered nowadays the main cause of functional impairment [45], the available literature in this field is poor. Real-world functioning in MDD can be divided into two distinct domains: (1) adaptive behavior, which includes activities necessary for everyday living (self-care, vocation, and household maintenance); and (2) interpersonal behavior, which represents the ability to initiate and maintain social contacts [1,46], with an important gap between competence and real-world actual performance, possibly mediated by negative self-efficacy beliefs [47]. During the years, authors evaluated the impact of MDD on several aspects of functioning and on QoL, concluding that depressive and cognitive symptoms affect functioning in real-life, in direct proportion to increase of severity [48,49,50], and that a worse quality of life and functioning at baseline is significantly related to poor treatment outcome [48,49,50,51,52]. Additionally, in this disorder, recent studies focused on investigating the impact of cognitive symptoms (i.e., impairment of concentration and attention), which can hamper the functional performance [53,54,55,56,57,58,59].

Little is known about functional impairment associated with severe personality disorders (PD), and only a limited number of studies [60,61] have investigated these concepts in samples of patients with different diagnoses of PD [62,63,64]. The majority of the available studies focused mainly on borderline personality disorder (BPD), exploring the functional course of the disease [65,66,67,68,69,70]. Such papers found that the recovery of psychosocial functioning in BPD patients was less substantial than symptomatic improvement and that patients experience an important occupational and social impairment [68], similar to patients with MDD [71], which persists long after the symptomatic remission in a vast portion of patients [60]. However, these concepts, including engagement in meaningful vocation and relationships, are partly different from that of real-world functioning, which measures the actual performance of patients in the real world [66].

This observational investigation is aimed at identifying, in a sample of outpatients with a diagnosis of schizophrenia, bipolar disorder, major depressive disorder, or borderline personality disorder, whether socio-demographic and clinical variables may predict a change of the real-world functioning.

## 2. Materials and Methods

### 2.1. Subjects

The present study was conducted in 1019 outpatients who attended the “Struttura Complessa di Psichiatria Universitaria” of the Department of Neuroscience “Rita Levi Montalcini” of the University of Turin, Italy. Patients were enrolled between June 2015 and December 2021. Evaluation period was one year. The study was carried out in accordance with the Declaration of Helsinki of 1995 (as revised in Edinburgh in 2000) and was approved by the Local Ethical Committee (LREC; Protocol number: 0057625). For all patients, the written informed consent was collected prior to their participation and after a complete description of the study. The study scrupulously followed the rules on the handling of biomedical data (Council of the EU: Data Protection, 2015).

### 2.2. Inclusion and Exclusion Criteria

Participants met a diagnosis of: (1) major depressive disorder, or (2) bipolar disorder, or (3) schizophrenia, or (4) borderline personality disorder (according to DSM-5 criteria). Disorders were diagnosed by an expert clinician (P.B.) and were confirmed using the Structured Clinical Interview for DSM-5-Clinical Version (SCID-5-CV) and Personality Disorders (SCID-5-PD) [72,73].

Age of inclusion was between 18 and 60 years.

The exclusion criteria were lifetime diagnoses of delirium and/or neurocognitive disorders (major or mild).

### 2.3. Participants

Patients received standard care provided in community mental health centers in Italy, including pharmacotherapy, in accordance with the international guidelines, clinical monitoring with psychiatric visits at least monthly, and psychotherapy for selected patients.

Medications commonly prescribed in clinical practice in Italy include antidepressants (sertraline, paroxetine, escitalopram, and duloxetine), mood stabilizers (lithium salts, valproic acid, and lamotrigine), and antipsychotics (olanzapine, risperidone, aripiprazole, and quetiapine).

In patients with a diagnosis of MDD or BPD, interpersonal psychotherapy (IPT) was offered. Patients with MDD were treated with traditional IPT lasting 16 weeks [74], while patients with BPD were treated with the revised adaptation of IPT to BPD (IPT-BPD), lasting 10 months [75]. Patients with schizophrenia and bipolar disorder were not treated with psychotherapy in our study.

### 2.4. Assessment

Sociodemographic and illness-related characteristics were collected during the psychiatric visits, with a semi-structured interview at baseline (t0). Anamnestic reports were confirmed when possible, by family members or caregivers. Data were entered in a password-protected database.

Categorical variables included: gender, working (employed/not employed), marital status (with/without long-term relationships), and living status (independent/dependent) considered as indicators of attainment in functional milestones, as well as psychiatric anamnesis, suicide attempts, and actual treatment with antipsychotics and/or mood stabilizers and/or antidepressants and/or psychotherapy. Medication intake was recorded by directly asking the patients and caregivers. Age, education level, age at onset (first psychiatric visit), illness duration (years passed from the first psychiatric visit), and previous hospitalizations (voluntary or mandatory) were included as continuous variables.

Furthermore, patients were tested at baseline (t0) with the following evaluation tools.

Clinical Global Impression Scale (CGI) [76] has been used to evaluate the overall severity of the illness. It is a three-item clinician-rated instrument which consists of three different measures: severity of illness (CGI-S), global improvement (CGI-I), and efficacy index (CGI-E). For the purpose of this study, we considered the first item (CGI-S), that is a seven-point scale testing the severity of the illness at the time of the assessment. A higher score corresponds to a greater severity.

The attitude towards psychiatric medications was rated using the Drug Attitude Inventory-10 (DAI-10) [77]. It is a self-report scale (short-version of the DAI-30). The scoring ranges between −10 and +10, where a total score greater than zero indicates a positive attitude toward medications, while a total score of less than zero indicates a negative attitude.

The personal satisfaction with different aspects of daily living was evaluated with the Satisfaction Profile (SAT-P) [78]. It is a self-evaluation instrument including 32 scales, which can be considered as subjective indicators of quality of life (psychological, physical, psychophysical, relational, and work-related). For this study, we used the “factors-related score”: for each item the patient indicated his satisfaction in the last month on a scale ranging from “extremely dissatisfied” (0) to “extremely satisfied” (100). In accordance with our choice, some authors reported that self-reporting measures of quality of life are more adequate than clinician-evaluating measures [1,79,80].

Personal and Social Performance (PSP) [81] was used to assess functioning. The PSP is a clinician-rated scale specifically designed to evaluate real-world functioning in patients during the course of treatment. This instrument comes from the Social and Occupational Functioning Assessment Scale—SOFAS [82] and measures: (1) socially useful activities, (2) personal and social relationships, (3) self-care, and (4) disturbing and aggressive behaviors. Each area is rated on a six-point scale from “absent” to “very severe” impairment. Out of the rating on the four subdimensions, a total score between 1 and 100 can be established, in which a higher score corresponds to a better functioning. The PSP scale has been re-administered to all patients after 12 months of standard treatment (t1).

### 2.5. Statistical Analyses

Statistical analyses were performed with the Statistical Package for the Social Sciences, SPSS, version 28 for Windows (SPSS, Chicago, IL, USA).

First, we calculated linear regression for continuous variables and performed Chi-square test for categorical variables. Linear regression between change of PSP score and the continuous variables (age, age of illness onset, level of education, illness duration, drug attitude (DAI-10 score), level of global symptoms (CGI-S score), and subjective perception of quality of life (SAT-P score)) was performed. Chi-square test was performed for the categorical variables (gender, working, marital status and independent living, positive psychiatric anamnesis, record of suicide attempts, use of antipsychotics, use of antidepressants, use of mood stabilizers, and ongoing psychotherapies).

Then, all socio-demographic and clinical variables that reached statistical significance at the univariate analysis (*p* ≤ 0.05) were included in a multiple regression analysis (stepwise backward). The difference of PSP score between 12 months (t1) and baseline (t0) (ΔPSP) was used as the dependent variable.

Statistical analyses were performed in the whole sample of patients and then in each subsample of patients with diagnosis of schizophrenia, major depressive disorder, bipolar disorder, and borderline personality disorder. Significance level was *p* ≤ 0.05.

## 3. Results

Of the 1019 outpatients in our sample, 17.9% had a diagnosis of schizophrenia, 42.4% a diagnosis of major depressive disorder, 21.5% a diagnosis of bipolar disorder, and 18.1% a diagnosis of borderline personality disorder. There were 376 males (36.9%), the mean age (mean ± SD) was 51.01 ± 15.32 years, and the mean level of education (mean ± SD) was 11.12 ± 4.21 years. Patients showed a moderate impairment in real-world functioning, with a PSP total score (mean ± SD) = 64.32 ± 10.75, with a lower level for patients with SZ (57.94 ± 10.57). The mean rate of hospitalization in our sample was 0.73 ± 2.5. The mean number of days in hospital was 6.2 ± 3.4.

The demographic and clinical features of the total sample and of the four diagnostic groups are presented in Table 1.

In the whole sample of 1019 patients, continuous variables that were found to be significantly related to the change of PSP score were: age of illness onset (Bst = 0.13; *p* = 0.001); duration of illness (Bst = −0.09; *p* = 0.008); number of hospitalizations, both voluntary (Bst = –0.07; *p* = 0.03) and mandatory (Bst = –0.08; *p* = 0.008); PSP score at T0 (Bst = 0.40; *p* = 0.001); CGI-S score (Bst r = 0.11; *p* = 0.001); DAI-10 score (Bst = 0.21; *p* = 0.001), and SAT-P score (Bst r = 0.61; *p* = 0.001). Categorical variables with a significant value of Chi-square test were: medications with antipsychotics (χ^2^ = 109.40; *p* = 0.001), with mood stabilizers (χ^2^ = 93.43; *p* = 0.001), with antidepressants (χ^2^ = 106.83; *p* = 0.001); psychotherapy (χ^2^ = 236.69; *p* = 0.001); being employed (χ^2^ = 392.20; *p* = 0.001); having a stable relationship (χ^2^ = 334.20; *p* = 0.001); and living independently (χ^2^ = 282.81; *p* = 0.001). All these variables were associated with better values of ΔPSP. On the contrary, male gender (χ^2^ = 88.01; *p* = 0.001); positive psychiatric anamnesis (χ^2^ = 86.79; *p* = 0.001); and suicide attempts (χ^2^ = 122.53); *p* = 0.001) were associated with lower values of ΔPSP. All significant variables were included in the multiple regression analysis. Results showed that age of illness onset (*p* = 0.001); antipsychotics (*p* = 0.001), mood stabilizers (*p* = 0.001), and psychotherapy (*p* = 0.001); SAT-P score (*p* = 0.001); PSP score at T0 (*p* = 0.001); being employed (*p* = 0.001); having a stable relationship (*p* = 0.001); and living independently (*p* = 0.001) were independently related to the change of PSP score. A positive psychiatric anamnesis (*p* = 0.01), use of antidepressants (*p* = 0.025), and CGI-S score (*p* = 0.001) were independently, but inversely related.

In the group of patients with a diagnosis of SZ, the following continuous variables were found to be significant: PSP score at T0 (Bst = 0.37; *p* = 0.001), SAT-P score (Bst = 0.66; *p* = 0.001), DAI-10 score (Bst = 0.77; *p* = 0.001), and CGI-S score (Bst = 0.23; *p* = 0.003). Categorical variables with significant results of Chi-square test were: being employed (Bst = 59.07; *p* = 0.001), having stable relationships (Bst = 32.70; *p* = 0.02), and living independently (Bst = 82.14; *p* = 0.001). These variables were associated with better values of ΔPSP. Male gender (Bst = 38.64; *p* = 0.005) presented lower values of ΔPSP. It has been found that SAT-P (*p* = 0.001), DAI-10 score (*p* = 0.001), PSP score at T0 (*p* = 0.001), working (*p* = 0.001), having stable relationships (*p* = 0.001), and living by themselves (*p* = 0.001) were independently related to the PSP change in the multiple regression.

In the group of patients with a diagnosis of MDD, we found significant results for the following continuous variables: age (Bst = –0.15; *p* = 0.001), age of illness onset (Bst = –0.10; *p* = 0.03), PSP score at T0 (Bst = 0.45; *p* = 0.001), CGI-S score (Bst = 0.24; *p* = 0.001), DAI-10 score (Bst = 0.27; *p* = 0.001), and SAT-P score (Bst = 0.79; *p* = 0.001). Categorical variables with significant results of Chi-square test were: working (χ^2^ = 175.24; *p* = 0.001), having relationships (χ^2^ = 172.86; *p* = 0.001), living independently (χ^2^ = 116.14; *p* = 0.001), medications with antidepressants (χ^2^ = 129.16; *p* = 0.001), and psychotherapy (χ^2^ = 207.95; *p* = 0.001). These factors were related to higher values of ΔPSP. On the contrary, male gender (χ^2^ = 61.16; *p* = 0.001), positive psychiatric anamnesis (χ^2^ = 67.59; *p* = 0.001), and record of suicide attempts (χ^2^ = 97.78; *p* = 0.001) were associated with less favorable values of ΔPSP. Variables significantly and independently related to PSP change in the multiple regression analysis were: PSP score at T0 (*p* = 0.001), SAT-P score (*p* = 0.001), having a stable relationship (*p* = 0.001), being employed (*p* = 0.001), and psychotherapy (*p* = 0.001). Variables inversely related to PSP change were: medications with antidepressants (*p* = 0.001), and CGI-S (*p* = 0.006).

In the group of patients with a diagnosis of BD, the continuous variables significantly related with PSP change were: age of illness onset (Bst = 0.25; *p* = 0.001), duration of illness (Bst = –0.22; *p* = 0.003), education level (Bst = –0.19; *p* = 0.01), PSP score at T0 (Bst = 0.5; *p* = 0.001), CGI-S score (Bst = 0.48; *p* = 0.001), DAI-10 score (Bst = 0.40; *p* = 0.001), and SAT-P score (Bst = 0.73; *p* = 0.001). Significant dichotomic variables at Chi-square test were: working (χ^2^ = 121.02; *p* = 0.001), having relationships (χ^2^ = 140.24; *p* = 0.001), living independently (χ^2^ = 92.76; *p* = 0.001), medications with antidepressants (χ^2^ = 44.47; *p* = 0.01), mood stabilizers (χ^2^ = 73.05; *p* = 0.001), and antipsychotics (χ^2^ = 45.63; *p* = 0.007). These factors had a positive association with ΔPSP, while a negative association was found for male gender (χ^2^ = 55.71; *p* = 0.001), positive psychiatric anamnesis (χ^2^ = 60.70; *p* = 0.001), and suicide attempts (χ^2^ = 52.34; *p* = 0.001). In the multiple regression analysis variables significantly and independently related to PSP change were: SAT-P score (*p* = 0.001); having a stable relationship (*p* = 0.001); working (*p* = 0.002), age at illness onset (*p* = 0.01), and medications with antipsychotics (*p* = 0.02). A positive psychiatric anamnesis (*p* = 0.003) was inversely related to PSP improvement.

In conclusion, in the group of BPD patients, we found significant results for the following continuous variables: age of illness onset (Bst = 0.18; *p* = 0.03), years of education (Bst = 0.17; *p* = 0.02), PSP score at T0 (Bst = 0.52; *p* = 0.001), CGI-S score (Bst = 0.36; *p* = 0.001), DAI-10 score (Bst = –0.38; *p* = 0.001), and SAT-P score (Bst = –0.05; *p* = 0.05). Regarding dichotomic variables, we found significant values of Chi-square test for: working (χ^2^ = 127.81; *p* = 0.001), having relationships (χ^2^ = 120.23; *p* = 0.001), living independently (χ^2^ = 73.09; *p* = 0.001), antidepressants (χ^2^ = 66.61; *p* = 0.001), mood stabilizers (χ^2^ = 40.47; *p* = 0.007), antipsychotics (χ^2^ = 48.82; *p* = 0.001), and psychotherapy (χ^2^ = 160.02; *p* = 0.001). These factors were associated with more favorable values of ΔPSP. Male gender (χ2 = 51.14; *p* = 0.001) and suicide attempts (χ^2^ = 43.73; *p* = 0.003) had a negative association with ΔPSP. Variables significantly and independently related to PSP change with the multiple regression analysis were PSP score at T0 (*p* = 0.001), SAT-P score (*p* = 0.001), being employed (*p* = 0.001), having relationships (*p* = 0.001), living independently (*p* = 0.005), use of mood stabilizers (*p* = 0.002), antipsychotics (*p* = 0.04), and psychotherapy (*p* = 0.03). Variables inversely related to PSP improvement were CGI-S (*p* = 0.001) and record of suicide attempts (*p* = 0.01).

Table 2 summarizes significant results of multiple regression analysis of the total sample and of the four diagnostic subgroups.

## 4. Discussion

Impaired real-world functioning, represented by different degrees of difficulties in attaining life milestones (employment, independent living, and a stable interpersonal relationship), is a feature shared by several psychiatric disorders. Among factors that may influence the global functioning in the real world, quality of life, treatment adherence, and severity of psychiatric symptoms are widely studied [13,14,83,84]. Our study was aimed to evaluate whether several socio-demographic, illness-related, and clinical variables can predict the change of functioning in real-world outpatients with different psychiatric disorders at 12 months follow-up.

Several previous investigations focused on predictive factors of functional outcomes in a single baseline evaluation or by comparing patients to healthy controls. However, few recent studies investigate the relationship between these factors and the change of real-world functioning over an observation period in patients with SZ [17,18,20] and in patients with MDD [49,50]. In this context, our results are only partially comparable to the available results.

It is not surprising that the main finding reported of the present study is that global real-world functioning at baseline, measured with the PSP scale, was one of the major determinants of the improvement of functioning after the follow-up period in all subgroups of patients other than BD, which confirms results of similar investigations [15,17,20]. It is rather difficult to explain why BD had a different behavior, that is, why a better functioning at baseline was not predictive of functional improvement at endpoint. We can only suggest that the dramatically unstable condition of BD with inconstant sequence of counter-polar episodes and free intervals is a serious obstacle to obtain a progressive modality of real-world functional improvement that can be predicted by initial functioning.

On the basis of literature data, the achievement of important milestones at baseline (such as having a stable work and relationship and living independently, which can be considered indices of good functioning) were found to be predictors of functioning amelioration at follow-up as well, even if in MDD and BD clusters not all milestones reached statistical significance in multiple regressions. In contrast with other studies, only Sylvia et al. [38] found that married or divorced patients with bipolar disorder experienced a worse functioning in comparison with singles or never married patients.

Other independent variables were significantly associated with functioning of patient in real-world, in particular the age of illness onset, the subjective perception of quality of life, and the type of treatments. Available data are in accordance with our findings indicating that patients with early-onset of mental illness (and thus a long-lasting disorder) have major deficits in functional outcomes [40,79,85,86,87,88,89], probably due to the fact that the onset of the disease occurs in a critical period, when specific markers of social status (housing independence, work abilities, and social relationships) and neurocognitive functions are still developing [90,91,92]. In the present study, we found a positive association between an older age of illness onset and a better functioning both in the total sample and in the subgroup of patients with BD. Literature findings are somewhat different, as this link is well established in SZ, as pointed out in a recent meta-analysis by deWinter et al. [93], while findings are controversial and received little attention in BD [94,95,96,97].

Consistent with our result showing that a better perception of quality of life by the patient was related to an improvement of real-world functioning in the whole sample of patients and in each diagnostic category, some authors found a strong relationship between patients’ functioning and quality of life, in terms of physical health status, psychological status and well-being, and interpersonal interaction [98,99]. In particular, the patients’ global life satisfaction [100], and better daily functioning were considered the main indicators of recovery and better quality of life [101,102]. Quality of life was found to be an important outcome indicator and predictor of symptomatic and functional improvement, especially in schizophrenia [102,103].

Moreover, in the total sample and in the subgroups of BD and BPD patients, the therapeutic intervention was identified as a predictor of good real-world functioning, in particular antipsychotics and mood stabilizers in the total sample and in the BPD group, and antipsychotics in the BD subsample. The importance of different kinds of therapies in recovering and returning to the premorbid functioning, or at least to a more efficient one, is well known and is supported by last years’ studies indicating the need of taking into account functional outcomes next to symptomatic ones in evaluating therapeutic effects in psychiatric disorders [55,104,105,106]. Adherence to drug therapy in the subgroup of patients with a diagnosis of schizophrenia resulted as a predictor of real-world functioning change rather than the type of medication. In fact, another important finding in our investigation is the role of attitude to pharmacotherapy, evaluated through the use of DAI-10, in predicting a change in daily functioning in SZ patients: a better attitude, and thus adherence to antipsychotic treatment, produced a positive effect on real-life functioning. Previous studies performed in schizophrenia showed similar findings [20,87,107,108,109,110]. Authors observed that functional performances and community integration correlated positively with subjective satisfaction with treatment and, in particular, adherence to antipsychotic medications, while lower adherence to medications predicted a poor psychosocial functioning.

A rather surprising finding of our research is the inverse association between the use of antidepressants and real-world functioning: in fact, in the total sample and in the MDD group, the use of antidepressants was found to be a predictor of worse functioning. Even though the majority of similar studies pointed out an opposite association [104], evidence is often controversial [88,111,112] and this particular correlation has been previously described at baseline [17]. It has been hypothesized that, in patients who need such therapies, the difficulties in social and work performances may be mediated by underlying dysfunctional cognitive processes related to the disease, such as communication, interpersonal interaction, and mentalization [55], which are in addition less responsive to current first-line antidepressant therapies [49,112,113] and could be considered as an important mediator of functional impairment [49,114]. However, this finding needs to be replicated and more clearly explained in its implications.

In agreement with the available literature, we observed a great PSP change in patients with MDD and with BPD that were treated with interpersonal psychotherapy (IPT). Positive effects of psychotherapeutic interventions on real-world functioning in psychiatric disorders have been shown by several previous investigations [115,116,117,118]. Most of the evidence concerns the improvement of social functioning in depressed patients who were treated with IPT [104,119,120,121,122]. Studies focused on the functional outcomes after psychotherapy in BPD are still scarce. In fact, the majority of investigations on psychotherapies specifically designed for BPD assessed the efficacy of interventions in terms of decrease of symptom severity [123,124] or evaluated functional outcomes of combined therapies after a long period of follow-up [68,125,126]. One recent review of ten studies by Zahediabghari et al. [70] concluded that specific psychotherapeutic interventions are useful to improve psychosocial functioning in patients with this personality disorder. Impairment of real-world functioning in BPD can be also attributable to interpersonal reactivity and instability of these patients [125]. Therefore, it is not surprising that IPT, a psychotherapeutic model specifically oriented to improve interpersonal relationship, produces an overall enhancement of daily functioning in the group of BPD subjects.

Another remarkable finding of our study pointed out that patients with a higher measure of illness severity (CGI-S score), index of a more severe degree of psychiatric symptoms, and a positive psychiatric anamnesis were associated with a decline in PSP score during the observation period of usual treatment. This significant relationship was found for both factors in the total sample, for CGI-S score in the MDD and in the BPD samples, and for positive psychiatric anamnesis in the BP sample. This result can be expected, as subjects with severe psychiatric diseases and a higher level of symptoms preventing them from achieving a good psychosocial functioning, have greater difficulty in reaching and maintaining real-word milestones. In a similar way, other investigations stated a significant relationship between severity of symptom domains and functional outcome in schizophrenia and mood disorders [20,49,112,127,128,129,130,131,132,133,134,135,136,137]. As for BPD, there is a considerable heterogeneity of results concerning relationships between PSP change and severity of global symptoms [69,138,139] that might depend on the noticeable variability of BPD symptoms producing fluctuating and unforeseeable effects on functioning in the real world.

with regards to the relationship between suicidal behaviors and everyday functioning, in our study we observed that BPD patients with a higher number of suicide attempts had a lower improvement in community functioning. A possible interpretation of this finding takes into account some evidence indicating that patients with a history of suicidal behaviors had lower abilities in domains of neurocognitive functioning [140]. Therefore, patients with reduced cognitive flexibility could have limited opportunity to improve their real-world daily-living skills.

### Strengths and Limitations

The main strength is the real-world setting and real-world data collection of the study. This method presents the advantage of avoiding the selection biases of randomized controlled designs. In addition, the study was conducted in a large cohort of patients with a longitudinal follow-up.

In light of the real-world design, this study has also some limitations: (1) some demographical and clinical data were collected retrospectively, (2) we do not use psychopathological or functioning evaluation instruments specific for each diagnostic category, but the same measures for all the sample. Thus, there may be additional clinical predictors that we have not considered and that may play a role in long-term functional outcome of patients, i.e., cognitive symptoms. In light of this, it might be useful to replicate this study in single diagnostic subgroups, using clinical variables and measures of functioning specific for each subgroup, and to assess the role of cognitive dysfunctions in functional outcome.

## 5. Conclusions

In conclusion, the results of this analysis demonstrated that several different socio-demographic and illness-related variables contribute to patients’ functioning in a real-world setting, besides psychopathology and severity of the disease. This means that it is important to put the focus towards factors beyond clinical symptoms (i.e., quality of life, attitude to treatments, access to incentives which allow patients to achieve important milestones), that represent important contributors to patient’s achievements in daily-life functioning. Further studies are required to replicate our findings. In fact, a more accurate and reliable knowledge of predictors of functional outcome in patients with main psychiatric disorders would be of great importance to design therapeutic interventions, for example interventions of psycho-rehabilitation, targeted to obtain specific goals in functional domains and perception of quality of life.

## Figures and Tables

**Table 1 jcm-11-04400-t001:** Demographic and clinical variables in the total group of outpatients and in the subgroups: with a diagnosis of schizophrenia, or major depressive disorder, or bipolar disorder, and or borderline personality disorder.

Variables	Total Sample *n* = 1019	SZ *n* = 183	MDD *n* = 432	BP *n* = 219	BPD *n* = 185
Age, mean ± SD	51.01 ± 15.32	45.64 ± 14.92	54.22 ± 15.67	53.71 ± 14.11	42.52 ± 16.38
Age at onset, mean ± SD	39.58 ± 19.03	27.93 ± 9.61	43.95 ± 17.22	44.02 ± 24.82	31.21 ± 17.09
Illness duration, mean ± SD	12.21 ± 11.93	16.72 ± 12.11	11.41 ± 12.98	12.68 ± 10.77	9.12 ± 9.98
Education, mean ± SD	11.12 ± 4.21	10.82 ± 3.61	11.01 ± 4.34	10.79 ± 4.32	12.45 ± 4.19
CGI-S, mean ± SD	4.23 ± 1.01	5.14 ± 0.79	3.56 ± 0.58	4.28 ± 0.92	5.04 ± 0.67
SAT-P, mean ± SD	61.11 ± 17.51	59.66 ± 18.87	67.67 ± 17.41	51.42 ± 16.25	48.34 ± 10.21
DAI-10, mean ± SD	2.01 ± 3.52	2.99 ± 4.22	4.31 ± 1.39	−1.14 ± 2.37	−2.61 ± 2.38
PSP, mean ± SD	64.32 ± 10.75	57.94 ± 10.57	65.89 ± 9.62	66.44 ± 11.61	65.05 ± 11.86
Male gender, *n* (%)	376 (36.90)	78 (42.72)	154 (35.65)	70 (31.96)	74 (40)
Positive psychiatric anamnesis, *n* (%)	782 (76.74)	160 (87.43)	330 (76.39)	158 (72.15)	134 (72.43)
Suicidal attempts, *n* (%)	78 (7.65)	8 (4.37)	34 (7.87)	14 (6.39)	22 (11.89)
Employment, *n* (%)	506 (49.65)	52 (28.41)	252 (58.33)	118 (53.88)	84 (45.40)
Stable relationship, *n* (%)	493 (48.38)	74 (40.44)	244 (56.48)	94 (42.92)	81 (43.78)
Independent living, *n* (%)	783 (76.84)	86 (46.99)	416 (96.30)	156 (71.23)	125 (67.57)
Antipsychotics, *n* (%)	346 (33.95)	183 (100)	22 (5.09)	60 (27.40)	81 (43.78)
Mood stabilizers, *n* (%)	331 (32.48)	19 (10.38)	18 (4.17)	174 (74.45)	120 (64.86)
Antidepressants, *n* (%)	537 (52.70)	33 (18.03)	426 (98.61)	62 (28.31)	16 (8.65)
Psychotherapy, *n* (%)	326 (32.00)	--	228 (52.78)	--	98 (52.97)
Diagnosis of SZ, *n* (%)	183 (17.96)	--	--	--	--
Diagnosis of MDD, *n* (%)	432 (42.39)	--	--	--	--
Diagnosis of BD, *n* (%)	219 (21.49)	--	--	--	--
Diagnosis of BPD, *n* (%)	185 (18.15)	--	--	--	--

SD = standard deviation; *n* = number; SZ = Schizophrenia; MDD = Major Depressive Disorder; BP = Bipolar Disorder; BPD = Borderline Personality Disorder; DAI-10 = Drug Attitude Inventory-10; CGI-S = Clinical Global Impression-Severity; SAT-P = Satisfaction Profile; PSP = Personal and Social Performance.

**Table 2 jcm-11-04400-t002:** Multiple regression analysis in the total group and in the subgroups with a diagnosis of schizophrenia, or major depressive disorder, or bipolar disorder, or borderline personality disorder. Dependent variable ΔPSP. * *p* ≤ 0.05 ; ** *p* ≤ 0.01.

Variables	Total Sample Bst/SE		SZ Bst/SE		MDD Bst/SE		BD Bst/SE		BPD Bst/SE	
Age at illness onset	0.08 **	0.01	--	--	--	--	0.12 **	0.02	--	--
Positive psychiatric anamnesis	−0.05 **	0.45	--	--	--	--	−0.14 **	1.33	--	--
Suicidal attempts	--	--	--	--	--	--	--	--	−0.11 **	0.78
Antipsychotics	0.12 **	0.42	--	--	--	--	0.11 *	0.88	0.10 *	0.65
Mood stabilizers	0.20 **	0.48	--	--	--	--	--	--	0.15 **	0.69
Antidepressants	−0.07 *	0.44	--	--	−0.09 **	0.98	--	--	--	--
Psychotherapy	0.13 **	0.35	--	--	0.17 **	0.40	--	--	0.16 *	0.96
PSP T0	0.30 **	0.02	0.30 **	0.03	0.28 **	0.03	--	--	0.52 **	0.04
CGI-S	−0.15 **	0.25	--	--	−0.08 **	0.35	--	--	−0.21 **	0.49
DAI-10	--	--	0.45 **	0.05	--	--	--	--	--	--
SAT-P	0.45 **	0.01	0.21 **	0.01	0.52 **	0.01	0.37 **	0.04	0.28 **	0.03
Employment	0.25 **	0.34	0.12 **	0.44	0.14 **	0.43	0.22 **	1.23	0.48 **	0.63
Stable relationship	0.13 **	0.33	0.16 **	0.40	0.15 **	0.40	0.26 **	1.11	0.25 **	0.63
Independent living status	0.12 **	0.42	0.30 **	0.48	--	--	--	--	0.13 **	0.70

SZ = Schizophrenia; MDD = Major Depressive Disorder; BP = Bipolar Disorder; BPD = Borderline Personality Disorder; DAI-10 = Drug Attitude Inventory-10; CGI–S = Clinical Global Impression–Severity; SAT-P = Satisfaction Profile; PSP = Personal and Social Performance; SE = Standard Error.

## Data Availability

Our local ethics committee does not allow us to make our sets of data available.

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
