# Peer review of "Real-World Functioning in Psychiatric Outpatients: Predictive Factors"

_jcm, 2022, doi:10.3390/jcm11154400_

Round 1

Reviewer 1 Report

This study discusses factors that improve outcomes for psychiatric outpatients. Few approaches have examined a variety of psychiatric disorders over a 12-month follow-up period as this paper does.

However, this manuscript does not describe psychiatric outpatient treatment in Italy, making it difficult to apply the results of this study to psychiatric treatment in other countries. The content of this study would be better understood if the rate of hospitalization and the number of hospital days in psychiatric care in Italy were presented.

Also, although interesting findings were made regarding antidepressants, we would like to see the antidepressants currently used in Italy presented.

Author Response

Psychiatric oupatient treatment in Italy was better described and some information were added in the text. Rate of hospitalization and number of hospital days were added in the results section. Antidepressants administered in our sample were specified. 

Reviewer 2 Report

Referee report on the paper entitled „Real-world functioning in psychiatric outpatients: predictive factors” by Bozzatello et al.

The paper deals with the predicting factors of the changing of personal and social performances in the case of patients with some mental diseases (SZ, MDD, BD, BPD). The sample size is quite large, which is to be welcomed.

The main finding of the paper that the changes of personal and social performances depend on some socio-cultural factors, (as education), on illness-related factors (for example the state at the baseline). These findings are not surprising at all. The statistical tests prove that the hypotheses of independency are rejected. Nevertheless, the largest correlations are with the state at base line and SAT-P-even these correlations are rather low-than medium. Table 2 presents the same. The effects of pairs (cross-effects) can also be investigated by regression. This may improve the findings.

In the case of dichotom variables it would be useful to conclude which case is more “dangerous” and which one is more favorable.

The authors list huge amounts of references, which are very time-consuming to check. It seems that lots of papers deal with the investigations of the different effects. This can be one among them.

Can the author explain why BD behaves differently from the others?

The paper contains lots of incompletions: keywords, author contributions, data has to be available, not only “will be available”…. Not published material contains the same tables which are in the paper. Why?

Author Response

Statistical analysis was performed again using linear regression for continuous variables. Significant results have not changed.

In the case of dichotomous variables, we indicated which case is more favorable.

Our aim was to provide a complete list of references, in rder to allow readers to compare our results with all previous investigations in our knowledge.

We tried to provide an explanation of the different behavior of BD patients based on clinical considerations.

Keywords and authors' contributions were added. Supplementary materials were not necessary in this study. Data are available in the tables. Our Ethical Committee does not allow us to make available further sets of data.

Reviewer 3 Report

The authors present an interesting topic. Nonetheless, some issues are presented, mainly the template was not followed:

-The abstract is over the 200 words, it has to be reduced

-The methods arise some questions the recruitment took place from 2015 to 2021, bu it is stated that the duration was one year. Do the authors the gathering of the data?

- The ethical approval number or code is provided, and the Biomedical and data law should be included.

- How was the data saved? how was the data obtained?

The only reference provided is "characteristics were collected using a semi-148 structured interview at baseline (to). "

-The results related to SZ include the Pearson's correlation for nominal variables (gender, being employed, having a stable relationship, and living independently ). I am assuming this is a mistake or it was transformed into a dummy for a model, please modify it since the following paragraphs are correct.

Author Response

Abstract has been shortened. Data were collected between 2015 and 2021. Period of observation for each case was 1 year. This point was made clear in the text.

Etical approval number was provided and reference of bio-medical data law was included.

Data were saved in a computerized database, protected by password.

We obtained demographical and clinical data during psychiatric visits. Data were confirmed by caregivers when possible. In addition to the semi-structured interview, a set of evaluation tools were used and were listed in the methods.

The results related to SZ paragraph were modified. 

Round 2

Reviewer 3 Report

The authors have modified the manuscript according to the comments previously provided. Nonetheless, the authors have not used the template from the journal, before publication,they must adapt it.